# Error Characteristic Analysis of Fluid Momentum Law Test Apparatus

**Song Yang, Xianyong Zhu \* and Hui Wang**

School of Mechanical and Aerospace Engineering, Jilin University, Changchun 130025, China; yangsongjlu@jlu.edu.cn (S.Y.); whui99@jlu.edu.cn (H.W.)
\* Correspondence: zhuxy@jlu.edu.cn; Tel.: +86-1361-430-4989

**Abstract:** The flat-plate momentum test bench is a widely used experimental device in the verification of the momentum law of fluid mechanics, and its error characteristics are of positive significance for theoretical research and engineering innovation and expansion. The SPH-FEM coupling algorithm and spectrum analysis method are used to calculate and analyze the displacement response and spectrum characteristics of the characteristic points of the sensor under different jet loads. Based on them, the cause, classification, law, scope, influence and control method of the measurement error of the system are discussed and analyzed with the application of the error theory and the lateral effect theory of strain gauges; combined with physical experiments, the relevant analysis methods and conclusions are verified. The results show that the measurement error of the system includes linear error and periodic error. Structural deformation in the direction of jet impact is the main source of linear error; linear error increases with the increase of jet loads. Meanwhile, periodic vibration in non-jet direction is the main cause of periodic error, and the periodic error decreases with the increase of jet loads.

**Keywords:** error analysis; momentum law; SPH-FEM coupling algorithm; fluid mechanics; spectral analysis

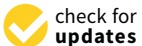



## 1. Introduction

Fluid mechanics is a basic science for studying fluid phenomena and related mechanical behaviors. Its application scopes cover aerospace engineering, mechanical engineering, power engineering, chemical engineering, biomedicine and other fields [1–3]. Fluid mechanics plays a broad and huge role in promoting scientific development, technological progress and industrial upgrading. The law of fluid momentum is one of the three basic laws of fluid mechanics as a basic theory explaining the relationship between fluid momentum change and force. The verification experiment of the law of constant flow of incompressible fluids, as the basic experiment to verify the theory of fluid momentum, is the core classic experiment of fluid mechanics [1,4,5]. The water jet impingement plate model method is a common experimental method to verify the law of fluid momentum. During the experiment, by changing the initial momentum of the jet, the force done on the plate and the momentum change of the jet were measured, and the momentum correction coefficient was solved to verify the law of conservation of momentum.

The phenomenon of jet impacting on solids, as the physical basis of the momentum law verification experiment, is also a natural phenomenon widely existing in nature [6]. Analyzing the error characteristics of the momentum law verification experiment through the jet impact phenomenon will help improve the scientificity, accuracy and reliability of the error analysis, and provide a new path for the exploration of the error generation mechanism. Jet impact has a wide range of engineering application backgrounds, and its applications cover aerospace engineering, power engineering, mechanical engineering and mining engineering [7–10]. The introduction of jet impact analysis into the fluid momentum law verification experiment will not only help improve the depth of momentum law

theoretical research, but also play a significantly positive role for improving the engineering innovation and application expansion based on the law of momentum [11–13].

Water jet impacting on a solid is a nonlinear dynamic process involving water hammer force, impact force (stagnation pressure), material deformation, damage and many other factors. It has the characteristics of large deformation, high strain rate, and transient processes that are difficult to observe and detect. [6,10]. There are many researches on the theory and numerical calculation methods of water jet impact at home and abroad, but there are few studies on the specific impact process and error characteristics of the momentum law verification experiment based on the principle of water jet impact on a flat plate. The author and his team have studied the impact of water jet on a flat plate. A preliminary study on the action process of the model has been carried out, and the relevant experimental data on the condition of the initial momentum of the jet equal to 15.74 Kg·m/s is obtained. The analysis shows that the irregular vibration of the sensor in the non-jet direction is the main reason for the change of measurement error [14].

This paper selects the momentum law experimental platform as the research object, establishes the numerical model of the water jet impingement momentum law experimental platform, uses the SPH-FEM coupling algorithm to analyze the jet impact on the plate, and obtains the jet load conditions at different speeds (different initial momentum). Based on the response characteristics of the sensor structure, the error characteristics of the momentum law experimental device are analyzed.

## 2. Experimental Device and Numerical Model

### 2.1. The Structure of Momentum Law Test Apparatus

The momentum law test apparatus consists of the following parts: ① water tank, ② pump-motor system, ③ flow metering device, ④ jet generator, ⑤ flat model, ⑥ base and ⑦ impact force measuring device (sensor and digital display meter), as shown in Figure 1a. The structure involved in the verification of the law of momentum, named as the core model, is shown in Figure 1b, which includes a circular flat plate, a resistive strain sensor, a base and a number of connecting parts. Their structural relation is as follows: the circular flat plate is fixed to the sensor through the central hole, the sensor is fixed to the base through bolts and the base is fixed on the base of the experimental device with glue. The circular plate and sensor are the core components. The function of circular plate is changing the jet flow to generate jet impact force, and the function of the sensor is measuring the impact force.

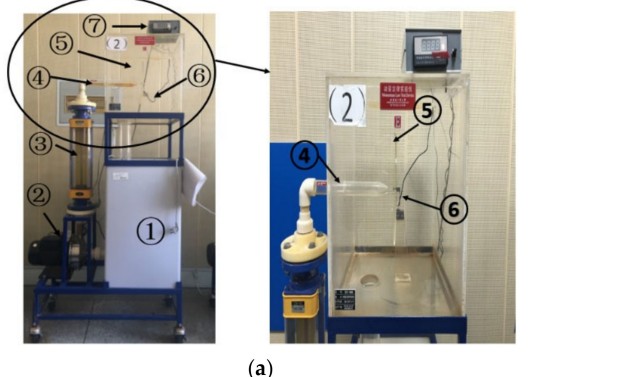
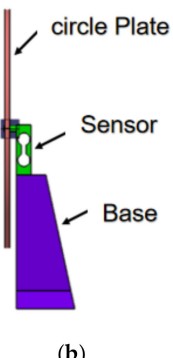

(a)                                                                 (b)

**Figure 1.** The structure graph of fluid momentum law test apparatus (**a**) the physical model of fluid momentum law test apparatus; (**b**) the core model of fluid momentum law test apparatus.

### 2.2. The Principle of SPH-FEM Coupling Algorithm

The process of water jet impacting solid structures is a typical fluid-solid coupling problem, which involves large deformation of the fluid. If the traditional Lagrange method is used to solve it, errors such as grid distortion are prone to occur, and the calculation

cannot be performed [14,15]. As a contact solution algorithm, SPH-FEM coupling algorithm is composed of two parts: SPH algorithm and FEM algorithm. SPH algorithm is a meshless algorithm, which has obvious advantages in solving problems such as large deformation, free surface flow and impact problems. In the solution of small deformation problems, there are disadvantages of insufficient calculation efficiency and accuracy. FEM algorithm is a traditional finite element method, which has obvious advantages in solving small deformation and vibration problems. According to the above, the two methods are combined and given full play to the advantages of the two algorithms to solve the jet impact problem, which includes the coexistence and coupling of physical characteristics such as large deformation, free flow, impact deformation and vibration [16].

### 2.2.1. The Principle of SPH Algorithm

The Smoothing Particle Hydrodynamics (SPH) method is a meshless, pure Lagrange method. The SPH method was first applied to fluid free surface flow by Monaghan in 1994. This method is currently widely used in the fields of fluid free motion, impact collision, explosion, soil mechanics, etc. [17–22].

The basic logic of the SPH algorithm is to replace the continuum fluid with a series of randomly distributed particles, and estimate the space function and its derivative of the N-S equation through the particle set and interpolation kernel function, thereby transforming the original N-S equation containing both time and space derivatives into an equation only containing the time derivative [17]. The SPH method includes two parts: integral representation and particle approximation. The kernel function is the core and basis of the integral representation part of the SPH algorithm. The kernel function is approximated by the integral field function, and its expression is shown in Formula (1). On this basis, the kernel approximation equation is used to re-approximate by particle by applying the values corresponding to adjacent particles in the local area to superimpose and replace the function and its derivative integral form, named as particle approximation, as shown in Formula (2). Finally, the particle approximation process of fluid mechanics equations is realized [23,24].

$$\{f(x)\} \approx \int_{\Omega} f(x')W(x - x', h)dx' \tag{1}$$

$$\langle f(x_i) \rangle = \sum_{j=1}^{N} \frac{m_j}{\rho_j} f(x_j) W(x_i - x_j, h) \tag{2}$$

In Formulas (1) and (2), $f$ is the field function, $\Omega$ is the calculation area, $x$ is the coordinate vector, $h$ is the smooth length, $W$ is the kernel function, $\rho$ is the density, and $m$ is the mass.

The SPH method is applied to the N-S equation, specifically as follows: mass conservation equation (Formula (3)), momentum conservation equation (Formula (4)) and energy conservation equation (Formula (5)); where $P_i$ represents the pressure of particle $i$, $v_i{}^\beta$ represents the velocity of particle $i$ in the $\beta$ direction, $x_i{}^\beta$ represents the coordinate of particle $i$ in the $\beta$ direction, $\sigma_i{}^{\alpha\beta}$, $\varepsilon_i{}^{\alpha\beta}$ represent the stress and strain tensor of the $i_{-th}$ particle, and $\mu$ represents the fluid viscosity coefficient, N represents the total number of particles in the smooth length range, $v_{ij}{}^\beta = v_i{}^\beta - v_j{}^\beta$ represents the component of the difference between the velocity of the two particles $i$ and $j$ in the $\beta$ direction [24].

$$\{f(x)\} \approx \int_{\Omega} f(x')W(x - x', h)dx' \tag{3}$$

$$\frac{dv_i^\alpha}{dt} = \sum_{j=1}^{N} m_j \left( \frac{\sigma_i^{\alpha\beta}}{\rho_i^2} + \frac{\sigma_j^{\alpha\beta}}{\rho_j^2} \right) \frac{\partial W_{ij}}{\partial x_i^\beta} \tag{4}$$

$$\frac{de_i}{dt} = \frac{1}{2}\sum_{j=1}^{N} m_j \left( \frac{p_i}{\rho_i^2} + \frac{p_j}{\rho_j^2} \right) v_{ij}^{\beta} \frac{\partial W_{ij}}{\partial x_i^{\beta}} + \frac{\mu_i}{2\rho_i}\varepsilon_i^{\alpha\beta}\varepsilon_j^{\alpha\beta} \tag{5}$$

The advantage of SPH algorithm to deal with the problem of fluid-solid impact is that its convection term is directly simulated by the movement of particles, which completely eliminates the problem of numerical divergence on the free interface, and ensures the clear and accurate tracking of the free interface; the mesh-free feature also avoids the net problems caused by lattice distortion and reconstruction.

### 2.2.2. SPH-FEM Coupling Method

In the coupling method of SPH and FEM, SPH particles are regarded as special nodal elements, and the force of the particles is applied to the surface of the finite element through the contact penalty function method, and the contact type is point-surface contact. In contact, SPH particles are the slave nodes and FEM elements are the main contact surfaces. The penetration state of SPH particles and the surface of the finite element needs to be checked in each time step. If the state is penetration, a penalty function is applied. If there is no penetration, nothing is performed [14,18].

### 2.3. Numerical Model and Parameters

In the process of experimental investigation, a water jet is generated by the jet generator (④) and the water jet impacts the flat model (⑤) under the action of inertia. During this process, the water jet is greatly deformed under the block of the flat model, and the momentum of the jet changes, forming a jet impact force. The flat model produces displacement and deformation under the action of the jet impact force. This process is a typical water jet impacting solid. The SPH-FEM method is suitable for the numerical solution of this experiment. During the experiment, the verification of the law of fluid momentum under different jet velocity states was achieved by changing the flow of the system. Combining the actual operation of the experiment, this article intends to analyze the characteristics of the jet impact process and the momentum law test apparatus under five common flow conditions, as shown in Table 1. The jet generator of the experiment platform produces a circular cross-section equal-diameter water jet with a diameter of 8 mm. The jet impact position is the center of the plate, the target distance is 2 mm, and gravity is ignored; the impact calculation time is set to 30 ms.

**Table 1.** The distribution of jet velocity.

| No | Flow Rate (m³/h) | Speed of Jet (m/s) | Initial Momentum of Jet (kg·m/s) | Increment of Jet (kg·m/s) |
|----|----|----|----|----|
| 1 | 2.2 | 12.16 | 7.43 | 0 |
| 2 | 2.4 | 13.26 | 8.84 | 1.41 |
| 3 | 2.6 | 14.37 | 10.38 | 1.54 |
| 4 | 2.8 | 15.48 | 12.04 | 1.66 |
| 5 | 3.0 | 16.58 | 13.82 | 1.78 |

The numerical model of the experimental device is shown in Figure 2. The material and element properties of each part of the model are shown in Table 2. The basic properties of the materials PMMA (Polymethyl-methacrylate, abbreviated as PMMA) and STEEL [25] are shown in Table 3; the EOS of the jet (WATER) equation of state selects US-UP, and the specific parameters are also shown in Table 3. According to the experimental conditions, the classical formula of the momentum theorem is used to calculate that the jet impact force is less than 20N, and the materials PMMA and STEEL of the structure are in the elastic deformation zone without plastic deformation.

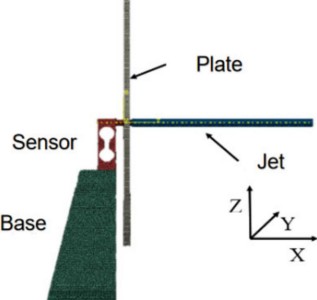

**Figure 2.** The numerical simulation graph of jet impact model.

**Table 2.** The categories of material and element of parts.

| No | Name | Material | Element Type | Element Size (mm) |
|----|------|----------|--------------|-------------------|
| 1 | Base | PMMA | SOLID | 2 |
| 2 | Sensor | STEEL | SOLID | 1 |
| 3 | Flat plate | PMMA | SOLID | 5 |
| 4 | Jet | Water | Particle | 1 |

**Table 3.** The properties of material PMMA, STEEL and WATER.

| Material Name | Density T/mm3 | Elasticity Modulus MPa | Poisson Ratio |
|---------------|---------------|------------------------|---------------|
| PMMA | $1.18 \times 10^{-9}$ | 5000 | 0.25 |
| STEEL | $7.89 \times 10^{-9}$ | $2.1 \times 10^5$ | 0.3 |
| | Density T/mm3 | Sound velocity $C_0$ mm/s | Viscosity MPa·s |
| WATER | $1 \times 10^{-9}$ | $1.45 \times 10^6$ | $1 \times 10^{-9}$ |

## 3. Analysis of Sensor Displacement Response Characteristics

The preliminary research foundation shows that the displacement response characteristics of the sensor structure can be used as the basis for error characteristic analysis [14]. In the momentum law experimental device, the sensor is arranged in a cantilever beam structure. With reference to the existing research results, the midpoint C on the side wall of the sensor is selected as the characteristic point of the displacement response characteristic of the sensor, referred to as the characteristic point C of the sensor, as shown in Figure 3 [14].

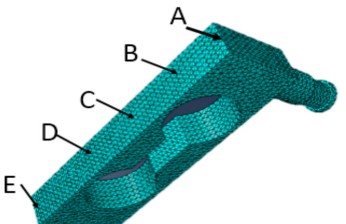

**Figure 3.** The distribution graph of observation points on sensor.

The displacement response of the sensor characteristic point C in the X, Y and Z directions under the action of the jet impact is shown in Figures 4–7, respectively. It can be seen from Figures 4–7, under the impact of different momentum jets, the sensor is displaced in the X direction (jet impact direction), Y direction and Z direction, for further analysis, a frequency spectrum analysis is performed on the displacement in the y direction, as

shown in Figure 6. The displacement amplitudes in the X direction and Y direction are larger than that of the Z direction. It shows that the external force that the sensor suffered in the X and Y directions is relatively large.

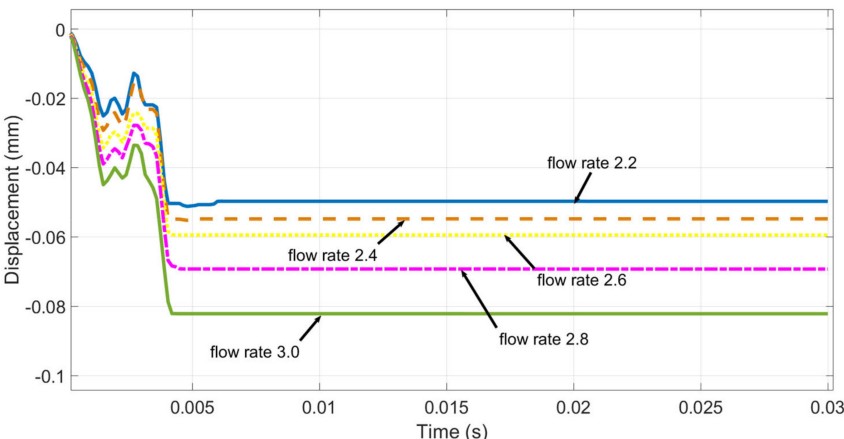

**Figure 4.** Displacement changing graph of sensor characteristic point C in X direction under the impacts of different momentum jets.

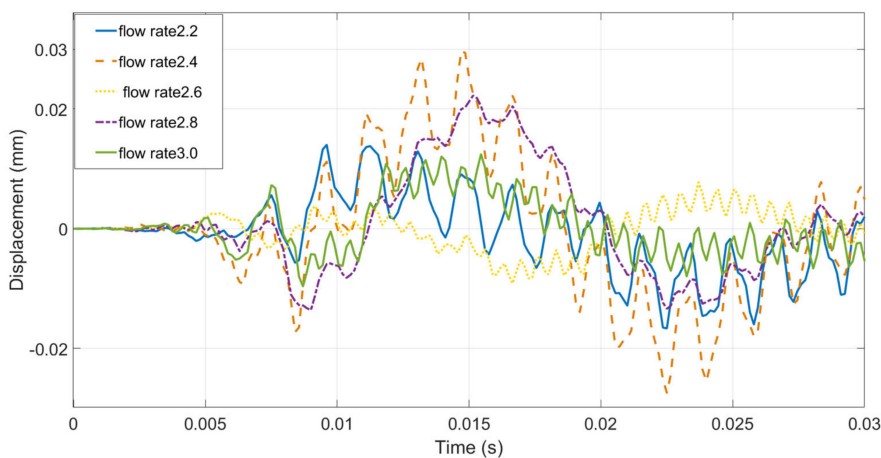

**Figure 5.** Displacement changing graph of sensor characteristic point C in Y direction under the impacts of different momentum jets.

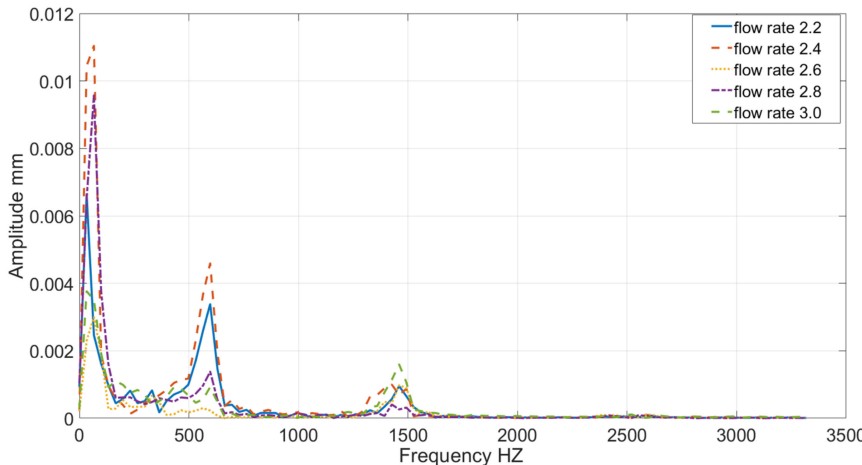

**Figure 6.** Spectral graph of displacements of sensor characteristic point C in Y direction under the impacts of different momentum jets.

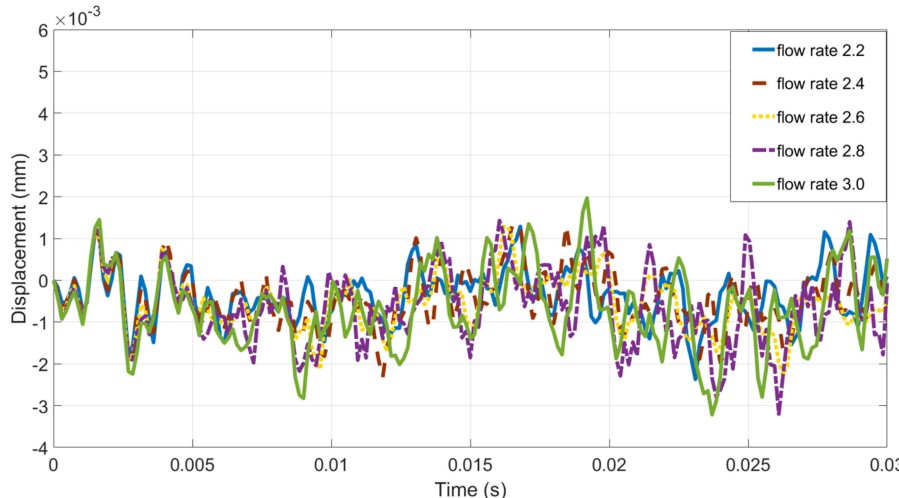

**Figure 7.** Displacement changing graph of sensor characteristic point C in the Z direction under the impacts of different momentum jets.

As Figure 4 shows, the impact force generated by the change of the jet flow momentum acts on the sensor in the X direction, which causes the characteristic point C to be displaced. The displacement curve of characteristic point C in the X direction becomes stable after oscillation adjustment. Under the action of different momentum jets, the oscillation law of characteristic point C is similar, but the amplitude and adjustment time are different. The basic rules are as follows: the greater the initial momentum of the jet, the greater displacement amplitude of the characteristic point C. The greater the jet flow increment, the greater the displacement increment of the characteristic point C, the longer the oscillation stability adjustment time, and all the oscillation adjustment time is less than 5 ms, which indicates that the force $F_{sx}$ of the sensor in the X direction become stable after a period of adjustment.

As Figure 5 shows, the characteristic point C is subjected to external force in the Y direction, resulting in displacement. The displacement curve of the characteristic point C in the Y direction has a small change in the time range of [0 ms, 5 ms]. After 5 ms, the vibration amplitude gradually increases, and the oscillation continues without a stable trend. This indicates that under the impact of the jet, the sensor first moves in X direction, after the adjustment in the X direction is stable, the sensor in Y direction starts to perform a large oscillation adjustment. Under the impact of different momentum jets, there is a certain difference in the displacement response trend of the sensor characteristic point C. There is no basic law that the greater the jet momentum, the greater the displacement of the characteristic point C in Y direction. In order to further explore its response law, frequency spectrum analysis is performed on the displacement response of the characteristic point C in Y direction, as shown in Figure 6. Under the impact of different momentum jets, the vibration frequencies of the sensor in the Y direction is similar, and the amplitude is different. The displacement vibration signal in the Y direction is a periodic signal, according to the Fourier transform theory, the periodic signal can be decomposed into a signal group, which is mainly composed of a 0-frequency signal and a set of periodic signals with different frequencies. In this article, the numerical sampling frequency is equal to 6666.7 Hz, the displacement vibration signal in the Y direction could be composed of four signals, including a 0-frequency signal and three sinusoidal periodic signals. The frequency of the periodic signal is distributed in the range of [33,66] Hz, [560,590] Hz and [1430,1460] Hz on the conditions of different flow rates. Taking into account the numerical calculation error and the relatively small range of the frequency distribution interval comparing to sampling frequency, to simplify analysis and calculation, it can be assumed considering that these three periodic signals are the fixed frequency. Take the middle of the frequency range and round up. Then, the frequencies of the three signals are 50 Hz, 580 Hz and 1450 Hz,

respectively. As the frequency increases, their amplitudes decrease in turn. The vibration amplitude is the largest when the jet flow rate is 2.4 m$^3$/h. Comparing Figures 4 and 5, it can be shown that the displacement of the sensor in the X direction is significantly greater than the displacement in the Y direction, which indicates that the force $F_{sx}$ of the sensor in the X direction is greater than $F_{sy}$.

As Figure 7 shows, under the action of the jet impact force, the characteristic point C of the sensor is displaced in the Z direction, and the displacement curve continues to oscillate. Under the same jet effect, the displacement amplitude in the Z direction is much smaller than the X and Y directions. Under the impact of different momentum jets, the displacements changing trend of the sensor characteristic point C are similar in time [0,5] ms; after more than 5 ms, the displacement changing trends are slightly different. The main reason for this situation is that the displacements and deformation of the sensor in the X and Y directions cause the change of displacement in the Z direction.

From the above analysis of the displacement response characteristics of the sensor characteristic point C under the impact of different momentum jets, it can be shown that under the action of the jet impact force, the sensor first moves in the jet direction (X direction), and tends to be stable. The greater the jet momentum, the greater the displacement in the X direction, and the longer the oscillation adjustment time. The sensor is subjected by non-constant external force in the Y direction to produce continuous vibration, and the vibration in the Y direction is greatly enhanced after the vibration in X direction enters the stable period. The vibration in the Y direction can be decomposed into a 0-frequency signal and three fixed-frequency periodic vibration signals. Under the impact of different momentum jets, the vibration frequency is equal and the vibration amplitude has a certain difference. The displacement of the sensor characteristic point C in the Z direction is mainly the result of sensor's displacements changing in the X and Y directions, and its displacement amplitude is much smaller than in the X and Y directions. It can be inferred that under the impact of the jet, the sensor first moves and deforms along the jet direction (X direction) and performs oscillation adjustment. After the jet direction (X direction) is finished and stabilized, periodic vibration occurs in the Y direction. The vibration in the Y direction is the main measurement error source.

## 4. Error Characteristic Analysis

The analysis in Section 3 shows that, under the impact of the jet, the sensor vibrates in the three directions of X, Y and Z. Combined with the relevant research that has been carried out, the transverse deformation caused by the irregular vibration is the main measured error source of resistance strain sensor. Considering the measurement principle and error theory of the resistance strain sensor, the relative displacement of the characteristic point C of the sensor relative to the point E is taken as the deformation *s* of the strain gauge, and the jet impact force obtained by the sensor expressed as $F = f(s)$, which is monotone increasing function [14].

The achieved studies have shown [14] that the impact force value measured by the sensor consists of a fixed value and a variable value. The displacement in the X and Y directions, respectively, affect the fixed value and the variable value measured by the sensor. The basic form of the impact force measurement value is: $c(t) + g(t)$, where $c(t)$ represents a fixed value, $g(t)$ represents a variable value; $g(t) = A_0 + A_1 sin(w_1 t + \varphi_1) + A_2 sin(w_2 t + \varphi_2) + A_3 sin(w_3 t + \varphi_3)$; $A_0$, $A_1$, $A_2$, $A_3$ represent the amplitude of the 0-frequency signal and three period signals, respectively. Under the impact of different momentum jets, the relative displacement or deformation of the sensor characteristic point C in the X and Y directions are shown in Figures 8 and 9, respectively.

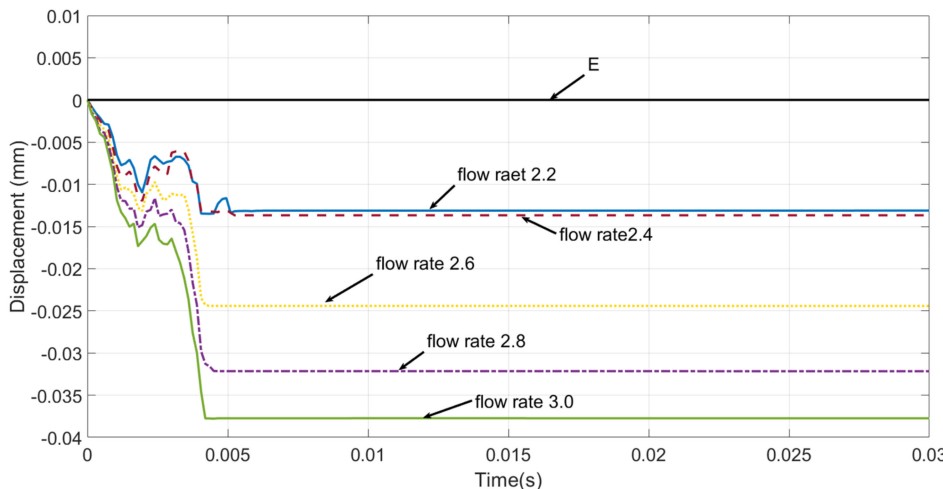

**Figure 8.** Relative displacement changing graph of sensor characteristic point under impacts of different momentum jets in the X direction.

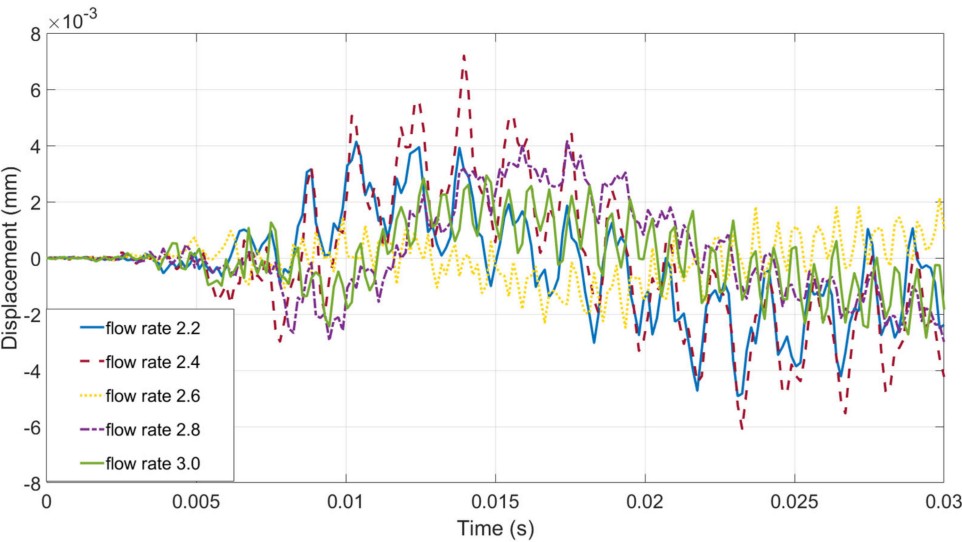

**Figure 9.** Relative displacement changing graph of sensor characteristic point under impacts of different momentum jets in the Y direction.

As shown in Figure 8, under the action of the jet impact load, the sensor strain gauge undergoes relative displacement and deformation in the X direction and tends to be stable after oscillation adjustment, that is, $c(t)$ tends to be a constant value after adjustment. The deformation $s_x$ of the sensor strain gage in the X direction and the jet flow are in a monotone increasing relationship, that is, the strain gauge deformation $s_x$ increases with the increase of the jet flow. However, the trend of the strain gauge deformation increment $\Delta s_x$ and the jet flow increment are different. The manifestation mode is that, as the jet flow increases, the influence of the jet flow increment on the increment of $\Delta s_x$ gradually decreases. The reason for this phenomenon is that the greater the jet flow, the greater the impact it produces. Since the sensor and the base structure are arranged in a cantilever beam structure, the greater the deformation, the greater the displacement of point E, and the resulting increase in the relative displacement of the two points C and E decreases. The result is that the deformation of the sensor strain gauge is reduced. Above all, it can be inferred that in the jet impact direction, a linear system error is caused by the deformation of the base; the larger the jet flow, the greater the linear system error caused by the deformation of the base. The linear error can be reduced by increasing the stiffness of the base [26].



As shown in Figures 9 and 10, the sensor strain gauge undergoes relative displacement and deformation in the Y direction under the action of the jet impact load. Within [0 ms, 5 ms], the displacement and deformation are basically 0. After 5 ms, it will continue to vibrate without stability. The vibration trend of strain gauge deformation is the same as the related analysis results in Section 3, and the amplitude of sensor strain gauge deformation $s_Y$ in the Y direction is still smaller than $s_x$, that is, $s_x > s_Y$.

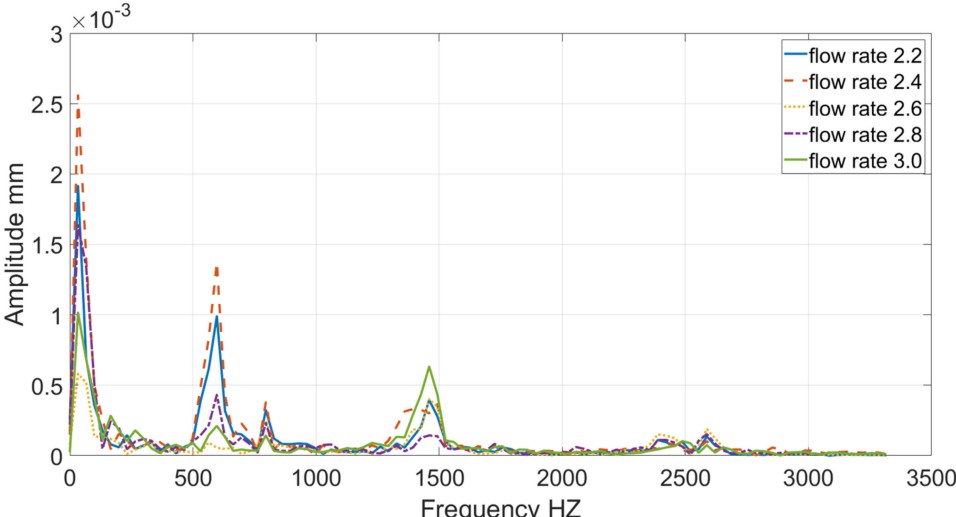

**Figure 10.** Relative displacement changing spectrum graph of sensor characteristic point under impacts of different momentum jets in the Y direction.

The continuous periodic vibration in the Y direction causes the deformation of the sensor strain gauge in the Y direction to change periodically, and the error caused by $s_Y$ is a periodic systematic error [26]. Comprehensively considering the vibration characteristics of the sensor, it is important to simplify the deformation $s_Y$ of the sensor strain gauge in the Y direction and accumulate the absolute value of each signal amplitude in the spectral characteristics of $s_Y$, as the maximum value of $s_Y$, that is, $s_{Ymax} = sum(A_0, A_1, A_2, A_3)$. According to the two-way response characteristics of strain gauges and the theory of lateral error, the calculation method of periodic system error caused by the sensor lateral effect can be expressed as [14,27]:

$$\delta = \frac{\varepsilon' - \varepsilon}{\varepsilon} \times 100\% = \frac{k\left(\frac{\varepsilon_y}{\varepsilon_x} + \mu_1\right)}{1 - \mu_1 k} \tag{6}$$

In Formula (6), $\delta$ represents relative error; $k$ represents the transverse sensitivity coefficient; $\varepsilon_x$ represents the strain in the X direction of the strain gauge; $\varepsilon_y$ represents the strain in the Y direction of the strain gauge; $\mu_1$ represents the Poisson's ratio of the strain gauge material.

The deformation $s_Y/s_X$ of the characteristic point C on the sensor is equivalent to $\varepsilon_x/\varepsilon_y$. After the sensor is stabilized in the jet direction (X), $s_X$ is a fixed value, and assumed as the strain gauge Poisson's ratio $\mu_1 = 0.3$. Take a different value for $k$ and estimating the error of the sensor under the different jets impacts the load, as shown in Table 4. It can be seen from Table 4 that the larger lateral sensitivity coefficient $k$ of the sensor, the larger measurement error. Under the same sensor parameter conditions, the larger jet flow, the smaller relative error.

**Table 4.** Sensor estimated error.

| Jet Flow Rate(m³/h) | $\mu_1$ | k | $\delta$ min% | $\delta$ max % |
|---|---|---|---|---|
| 2.2 | 0.3 | 0.01 | 0.30% | 0.69% |
| | | 0.02 | 0.60% | 1.39% |
| | | 0.03 | 0.91% | 2.10% |
| | | 0.04 | 1.21% | 2.80% |
| | | 0.05 | 1.52% | 3.52% |
| 2.4 | 0.3 | 0.01 | 0.30% | 0.81% |
| | | 0.02 | 0.60% | 1.63% |
| | | 0.03 | 0.91% | 2.45% |
| | | 0.04 | 1.21% | 3.28% |
| | | 0.05 | 1.52% | 4.11% |
| 2.6 | 0.3 | 0.01 | 0.30% | 0.39% |
| | | 0.02 | 0.60% | 0.77% |
| | | 0.03 | 0.91% | 1.17% |
| | | 0.04 | 1.21% | 1.56% |
| | | 0.05 | 1.52% | 1.95% |
| 2.8 | 0.3 | 0.01 | 0.30% | 0.43% |
| | | 0.02 | 0.60% | 0.86% |
| | | 0.03 | 0.91% | 1.30% |
| | | 0.04 | 1.21% | 1.73% |
| | | 0.05 | 1.52% | 2.17% |
| 3.0 | 0.3 | 0.01 | 0.30% | 0.38% |
| | | 0.02 | 0.60% | 0.77% |
| | | 0.03 | 0.91% | 1.16% |
| | | 0.04 | 1.21% | 1.55% |
| | | 0.05 | 1.52% | 1.95% |

The experiment on the apparatus is measured on the condition to the same as numerical parameters. The test data is shown in Table 5, which is similar to the estimated error (Table 4). Comparing the sensor estimated error (Table 4) and measured error (Table 5), it shows that most of measured errors are in the area that is ranged by estimated error. The result shows that the difference between the estimated error and the measured error is small, and the error analysis and estimation methods used in this article are effective. The reasons for the difference between the estimated error and the measured error are as follows [14,27–30]:

1. Ignoring the linear system error caused by the deformation of the jet direction base;
2. Error compensation inside the sensor;
3. Sensor measurement error caused by temperature and humidity;
4. The displacement response of point C cannot fully reflect the true change of the strain gauge;
5. The jet velocity changing caused by the pulsation of pump flow during the actual measurement;
6. The difference between the calculation model, material properties, etc. and the real system.

Comprehensive analysis shows that the experimental system error consists of two parts, namely, the linear system error caused by the jet impact direction (X direction) base deformation and the periodic system error caused by the sensor's Y direction periodic vibration and lateral effect. The linear error caused by the deformation of the base increases with the increase of the jet flow. The system error can be reduced by increasing the stiffness of the base; the measurement error caused by the lateral effect decreases with the increase of the jet, and the measurement error can be reduced by sensor correction.

**Table 5.** Experiment Error.

| Jet Flow Rate (m³/h) | Theoretical Impact Force *N* | Measured Impact Force *N* | Relative Error % |
|---|---|---|---|
| 2.2 | 7.43 | 7.51 | 1.08% |
| | | 7.36 | −0.94% |
| | | 7.59 | 2.15% |
| | | 7.45 | 0.27% |
| | | 7.49 | 0.81% |
| 2.4 | 8.84 | 8.95 | 1.24% |
| | | 9.10 | 2.94% |
| | | 8.93 | 1.02% |
| | | 8.85 | 0.11% |
| | | 8.82 | −0.23% |
| 2.6 | 10.38 | 10.40 | 0.19% |
| | | 10.45 | 0.67% |
| | | 10.35 | −0.29% |
| | | 10.27 | −1.06% |
| | | 10.30 | −0.77% |
| 2.8 | 12.04 | 12.15 | 0.91% |
| | | 12.20 | 1.33% |
| | | 11.95 | −0.75% |
| | | 12.10 | 0.50% |
| | | 12.23 | 1.58% |
| 3.0 | 13.82 | 13.95 | 0.94% |
| | | 13.75 | −0.51% |
| | | 13.80 | −0.14% |
| | | 13.90 | 0.58% |
| | | 14.00 | 1.30% |

## 5. Conclusions

This paper takes the jet impingement type momentum law experimental device as the research object, combined with the actual operation of the experiment, adopts the SPH-FEM numerical analysis method to analyze the displacement response characteristics and deformation characteristics of the characteristic point C of the sensor structure under different momentum jet loads. Based on these, the error cause, error classification, changing law, error scope and influence are analyzed and calculated, comparing with the experimental results, and the validity of the analysis and calculation method is verified. The conclusions obtained from the above analysis are as follows:

1.  Under the action of jet impact, the sensor structure vibrates and produces displacement and deformation. The jet direction (X direction) vibrates first and stabilizes after oscillation adjustment. After the vibration of the jet direction is stable, the non-jet Y direction starts periodic vibration;
2.  Among the vibrations in the non-jet direction, the vibration in the Y direction is the main source of error, and the vibration in the Z direction is mainly caused by the deformation caused by the vibration in the X and Y directions;
3.  The linear error of the system is mainly caused by the deformation of the base in the jet direction (X direction), and its size is positively related to the jet load. The linear error can be reduced by improving the stiffness of the base;
4.  The periodic error of the system is mainly caused by the periodic vibration in the Y direction and the lateral effect of the sensor. Its magnitude is negatively related to the jet load. The periodic error can be reduced by the method of sensor correction.

The analytical methods used and the conclusions obtained in this article, which have been partially applied to the experimental teaching of undergraduates to provide help for students to understand the causes of experimental errors.

**Author Contributions:** S.Y., X.Z. and H.W. conceived the conceptualization; S.Y. and H.W. designed the experiments; S.Y., X.Z. and H.W. performed the experiments, S.Y. and X.Z. analyzed the data; S.Y. wrote the paper. All authors have read and agreed to the published version of the manuscript.

**Funding:** Higher Education Research Project of Jilin Province (No. JGJX2020D18).

**Institutional Review Board Statement:** Not applicable.

**Informed Consent Statement:** Not applicable.

**Data Availability Statement:** Data is contained within the article.

**Conflicts of Interest:** The authors declare no conflict of interest.

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
