# Peer review of "Error Characteristic Analysis of Fluid Momentum Law Test Apparatus"

_applsci, doi:10.3390/app11010090_

Round 1

Reviewer 1 Report

A better quality of fig.1 a is required.

The data in tables 2 and 3 should be placed in a single table Fig.

6

The writing must be uniform (see Frequency)

Fig.

4-10 can be cut so that the white edges are removed and the figures become clearer

The conclusions are theoretical. A comparison between the measured results and the analytically calculated results would have been interesting.

The paper has a high scientific character and can be published after a minor revision

Author Response

Dear sir:

       The report , please see the attachment!

Reviewer 2 Report

The paper appears to present a comparison between SPH-FEM estimated and experimental error for the water jet impingement plate model and spectral analysis of oscillations.  The discussion suggests reasons for discrepancies between the estimated and experimental errors and solutions to the particular error types. 

Suggestions

  • Easily accessible (alternative) bibliographical citations.
  • Clearly stated connection between SPH-FEM model and experimental investigation
  • Supporting references for hypothesized error discrepanices (lines 313-319)
  • Experimental verification of proposed error reductions (lines 344-349)

Author Response

(The authors gave the same response as above.)
